# Silver Nanoparticles Embedded on Reduced Graphene Oxide@Copper Oxide Nanocomposite for High Performance Supercapacitor Applications

**DOI:** 10.3390/ma14175032

**Published:** 2021-09-03

**Authors:** Akhalakur Rahman Ansari, Sajid Ali Ansari, Nazish Parveen, Mohammad Omaish Ansari, Zurina Osman

**Affiliations:** 1Department of Physics, Faculty of Science, Universiti Malaya, Kuala Lumpur 50603, Malaysia; akhalakurkau@gmail.com; 2Center of Nanotechnology, King Abdulaziz University, Jeddah 21589, Saudi Arabia; moansari@kau.edu.sa; 3Department of Physics, College of Science, King Faisal University, P.O. Box 400, Al-Hofuf 31982, Saudi Arabia; sansari@kfu.edu.sa; 4Department of Chemistry, College of Science, King Faisal University, P.O. Box 380, Al-Hofuf 31982, Saudi Arabia; nislam@kfu.edu.sa; 5Centre for Ionics Universiti Malaya, Universiti Malaya, Kuala Lumpur 50603, Malaysia

**Keywords:** reduced graphene oxide, metal, metal oxide, electrochemical devices

## Abstract

In this work, silver (Ag) decorated reduced graphene oxide (rGO) coated with ultrafine CuO nanosheets (Ag-rGO@CuO) was prepared by the combination of a microwave-assisted hydrothermal route and a chemical methodology. The prepared Ag-rGO@CuO was characterized for its morphological features by field emission scanning electron microscopy and transmission electron microscopy while the structural characterization was performed by X-ray diffraction and Raman spectroscopy. Energy-dispersive X-ray analysis was undertaken to confirm the elemental composition. The electrochemical performance of prepared samples was studied by cyclic voltammetry and galvanostatic charge-discharge in a 2M KOH electrolyte solution. The CuO nanosheets provided excellent electrical conductivity and the rGO sheets provided a large surface area with good mesoporosity that increases electron and ion mobility during the redox process. Furthermore, the highly conductive Ag nanoparticles upon the rGO@CuO surface further enhanced electrochemical performance by providing extra channels for charge conduction. The ternary Ag-rGO@CuO nanocomposite shows a very high specific capacitance of 612.5 to 210 Fg^−1^ compared against rGO@CuO which has a specific capacitance of 375 to 87.5 Fg^−1^ and the CuO nanosheets with a specific capacitance of 113.75 to 87.5 Fg^−1^ at current densities 0.5 and 7 Ag^−1^, respectively.

## 1. Introduction

The depletion of fossil fuel reserves alongside the increase in environmental pollution is driven by a continual rise in world energy consumption. In addressing these issues, researchers are engaged in the development of renewable and cleaner energy resources such as solar photovoltaics and wind power. However, a major problem with these particular energy resources lies in their intermittency; a solar cell depends on sunlight and a wind turbine requires favorable wind conditions [1]. Hence, these energy resources cannot fulfill the energy demands of our society and so energy storage becomes a necessity. In recent years, energy storage devices such as batteries, capacitors and supercapacitors which are efficient, low cost, easy to manufacture and environmentally friendly have appeared on the market [2]. Batteries have high energy density and low power density whereas capacitors have high power density but low energy density meaning these energy storage devices have practical limitations. Supercapacitors provide a bridge between batteries and capacitors, in that they can have both very high energy density and power density, as well as stability after long cycle use [3]. Electrode materials play a very important role in supercapacitor performance. The ideal electrode should have a large specific surface area to store charge (high specific capacity), display high electrical conductivity to enable a high power density, exhibit high capacitance retention over many cycles and be made from environmentally sustainable materials [4]. Supercapacitors may be classed into two types according to the charge storage mechanism: either a non-faradaic electric double layer capacitor (EDLCs) or a faradaic pseudocapacitor [5,6]. In EDLCs the charge and discharge processes develop at the electrode-electrolyte interface and the capacitance is generated by ion adsorption from the electrolyte and driven by electrostatic force. However, in pseudocapacitors, the capacitance is generated by oxidation and reduction (redox) reactions between the electrolyte and the electrode surface. Carbon-based materials such as porous activated carbon, carbon nanotubes or graphene (GN) are ideal electrode materials for EDLCs owing to their large surface area and high activity [7] whereas conductive polymers and metal oxides are promising materials for electrodes in pseudocapacitors.

Transition metal oxides can have a variety of oxidation states which can increase redox reaction efficiency, making them suitable electrode materials for pseudocapacitors [8,9,10]. Metal oxides such as MnO_2_ [11], NiO [12], Co_2_O_4_ [13] and Vo_x_ [14] are promising candidate materials for supercapacitor applications. Recently copper oxide (CuO) has attracted attention due to its high abundance, low cost, lack of toxicity, good charge transport properties and its simple ability to be formed in differing shapes at a nanometer scale. As an electrode material, CuO has demonstrated high charge storage capacity for rechargeable Li-ion batteries [15,16,17,18] but there have been few investigations upon CuO as an electrode material in pseudocapacitors [19,20,21,22] owing to the rapid decay in capacitance due to poor electrical conductivity. These problems arise mainly from the CuO crystal structure that becomes distorted during the redox process. This drawback can be overcome by controlling the crystal structure’s size or incorporating the CuO in a composite material.

GN and reduced graphene oxide (rGO) are two-dimensional nanosheet structures of graphite. GN has been widely used as electrode material due to its large surface or specific area (2630 m^2^ g^−1^), high electrical conductivity (5 × 10^−3^ s cm^−1^) and high electrochemical activity [23,24,25,26]. By fully using its entire surface area GN or rGO might be able to achieve the theoretical maximum specific capacitance of 550 Fg^−1^ [27], but the reported specific capacitance found experimentally is much lower: 80–264 Fg^−1^ [23,27,28,29]. This decrease in capacitance may be due to the reduction of the electrochemically active surface area caused by the unavoidable aggregation or restacking of the rGO nanosheets. Therefore, by incorporating individual nanostructured material onto the sheets, and creating a well-defined hybrid nanocomposite, this aggregation or restacking might be avoided, and help preserve the electrochemical surface area. Recent studies have reported GN-based metal oxide nanostructures, such as TiO_2_ and Fe_2_O_3_ with GO thin sheets can reduce aggregation and increase the electrochemical surface area by creating more active sites during the redox process [30,31].

Noble metals such as platinum, ruthenium, gold, or silver can display properties such as high conductivity and electrochemical stability and, owing to their highly conductive nature, can enhance the redox process. Using silver (Ag) in combination with nanocomposites, electrical conductivity might be enhanced during the redox process by increasing the number of electron transfer channels. Electrode materials based on noble metal gold and silver oxides have been studied for pseudocapacitor applications [32,33]. Therefore, a synergetic effect can be achieved by decorating noble metals on the surface of a metal oxide and carbon matrix.

In this study, a ternary nanocomposite electrode material, Ag-rGO@CuO, was synthesized by several chemical routes. Here the transition metal oxide (CuO) reduces restacking of rGO nanosheets and may assist the Ag nanoparticles in providing extended channels and increase the rapid electron transfer. A ternary nanocomposite with these characteristics can increase electrochemical performance by improving cycling stability, specific capacity and rate capability. Synthesis of CuO, rGO@CuO and Ag-rGO@CuO was confirmed by several analytical techniques. The prepared pure metal oxide and its nanocomposites were used to form an electrode that demonstrated stable and excellent performance in potential supercapacitor applications.

## 2. Materials and Methods

### 2.1. Materials

Copper (II) nitrate trihydrate (CuN_2_O_6_.3H_2_O), silver nitrate AgNO_3_ (99.9%), potassium hydroxide (KOH), sodium borohydride (NaBH_4_), activated carbon (AC), anhydrous 1-methyl- 2-pyrrolidinone (NMP) (99.5%), absolute ethanol (C_2_H_6_O~99.8%), hydrogen peroxide (H_2_O_2_, 35%), potassium permanganate (KMnO_4_ > 99%) and hydrochloric acid (HCl~35%) were obtained from Sigma-Aldrich (Burlington, MA, USA). Graphite flakes were purchased from Asbury Inc. (Asbury, NJ, USA), polyvinylidene fluoride (PVdF) was obtained from Daejung Chemicals and Metal Co Ltd. (Gyeonggi, Korea). High-quality nickel foam (>99.99% purity) was bought from MTI Corporation (Richmond, CA, USA).

### 2.2. Synthesis of Electrode Materials

GO was synthesized from graphite flakes by a modified Hummer method. In this approach, 360 mL sulfuric acid (H_2_SO_4_) and 40 mL phosphoric acid (H_3_PO_4_) were poured into a 1 L glass beaker and mixed by magnetic stirring. Three grams of graphite flakes and potassium permanganate (18 g) were mixed ex-situ and added slowly to the acid mixture. The solution was stirred for 72 h at atmospheric temperature to obtain a dark green solution. This solution was poured over an ice cube of 600 mL water in a 2 L glass beaker. Hydrogen peroxide (H_2_O_2_) was added until the dark green solution turned yellow. The prepared yellow solution was centrifuged with deionized (DI) water several times to obtain a yellow gel of GO. Figure 1 shows the schematic diagram for the preparation of pure CuO with its binary and ternary nanocomposites.

Pure CuO nanosheets were synthesized by a microwave-assisted hydrothermal method. In this procedure, copper (II) nitrate trihydrate (CuN_2_O_6_·3H_2_O) and potassium hydroxide (KOH) were taken in a 1:20 atomic ratio and dissolved in 200 mL of DI water. The solution was stirred for 30 min to obtain an ultra-fine homogenous solution. The solution was placed in a microwave digester system with applied microwave power (750 W) for 10 min at 80 °C. The solution was cooled to room temperature and the resultant precipitate was collected. The precipitate dark brown color material was washed with DI water and absolute ethanol to remove the impurities. After further centrifugation, the dark brown product was annealed at 80 °C for 12 h to obtain CuO nanosheets.

To prepare rGO@CuO, a 2 mL GO solution at a concentration of 5 mg mL^−1^ was transferred dropwise onto 300 mg CuO nanosheets and mixed. The nanocomposite of GO@CuO was annealed at 400 °C for 4 h to reduce the GO into rGO and to get rGO@CuO composite.

To decorate the surface of rGO@CuO nanocomposite with Ag nanoparticles a chemical reduction method was employed. Solutions of 0.0010 M AgNO_3_ (aq) and 0.0020 M NaBH_4_ (aq) solution in DI water were separately prepared. A 90 mL portion of the sodium borohydride solution was added to a 500 mL Erlenmeyer flask and stirred for 20 min in an ice-cool atmosphere. 300 mg rGO@CuO of the nanocomposite was added and stirred for up to 10 min. Next, 30 mL of silver nitrate (AgNO_3_) was added dropwise by a burette. The precipitated dark yellow material was collected and annealed at 80 °C for 12 h to get Ag-rGO@CuO.

### 2.3. Characterization Techniques

To identify the phase and crystallinity of the prepared samples, X-ray powder diffraction (Rigaku, Ultima IV XRD, Tokyo, Japan) was employed. Field emission scanning electron microscopy (JEOL, JSM-7600F, FESEM, Tokyo, Japan) was used to study the surface morphology and elemental compositions. Transmission electron microscopy (JEOL, JSM, ARM-200F, HRTEM, Tokyo, Japan) and Raman spectroscopy (DXR Raman Microscope, Thermo Scientific fitted with a DXR 532 nm laser, Madison, WI, USA) were employed to examine the size, shape, and chemical composition of the nanocomposites, respectively. The electrochemical behavior of the electrode was studied by cyclic voltammetry (CV) and galvanostatic charge-discharge (GCD) using a Versa STAT 3 (AMETEK, Oak Ridge, TN, USA) electrochemical workstation.

### 2.4. Fabrication of Electrodes and Electrochemical Measurements

A chemically cleaned nickel foam was used to fabricate the electrode. For this, a 1 cm × 1 cm area of nickel foam was coated with slurry of the active material. The slurry of active material comprising 80 wt. % of the Ag-rGO@CuO nanocomposite, 10 wt. % of activated carbon (AC), and 10 wt.% of polyvinylidene fluoride (PVdF) was mixed in anhydrous 1-methyl-2-pyrrolidinone (NMP) and stirred homogeneously at ambient temperature for 12 h. The nickel foam was coated with the slurry and annealed at 90 °C for 12 h to dry. The loading of the active material was ~1.5 mg. The same method was used to prepare the working electrode for pure CuO nanosheets and the rGO@CuO nanocomposite. Electrochemical measurements on the electrode were performed using a three-electrode system in an aqueous 2M potassium hydroxide (KOH) electrolyte (Appendix A). In the assembled half-cell Ag/AgCl (3M KCL), platinum, and fabricated electrode served as reference electrode, counter electrode, and working electrode, respectively. Cyclic voltammetry (CV) was performed at different scan rates over a 0 to 0.5 V potential window. Galvanostatic charge-discharge (GCD) was measured at different current loads and across the same potential window.

## 3. Results and Discussion

### 3.1. X-ray Diffraction Studies

X-ray powder diffraction (XRD) was employed for phase identification and structural analysis of the prepared samples. Figure 2 shows the XRD pattern for pure CuO nanosheets, rGO@CuO, and Ag-rGO@CuO nanocomposites. In the pure CuO nanosheet pattern high-intensity peaks were located at 32.67°, 35.44°, 38.66°, 48.81°, 53.63°, 58.22°, 61.35°, 66.17°, 67.80° and 75.05°, and corresponded to the planes (110), (002), (111), (20−2), (020), (202), (11−3), (022), (113) and (004), respectively with a monoclinic phase [DB Card No. 01-080-1917]. The diffraction peaks of rGO@CuO nanocomposites are located at the same diffraction angle and plane as the CuO nanosheets. Additional peaks were also observed in the Ag-rGO@CuO sample at 38.04°, 43.16° and 64.17° attributed to the (111), (200), and (220) planes of the FCC phase of silver [DB Card No. 00-00-1167], respectively. The absence of a peak at ~26 2θ confirms the successful coating of rGO and the absence of any graphitic impurities [34]. The intensity peak of CuO nanosheets decreased in the Ag-rGO@CuO nanocomposite due to the introduction of Ag and rGO which cover some area of CuO sheets as well as due to the interactions between Ag, rGO and CuO. Similar results were also reported by Javed et al. [35], where peak intensity of Co_3_O_4_ decreased after the incorporation of MWCNT and Ag. Appendix A shows the XRD pattern of few layered and multilayered GO and rGO.

### 3.2. SEM, EDX and TEM Characterization

SEM images of the CuO nanosheets, rGO@CuO, Ag-rGO@CuO and EDX of the Ag-rGO@CuO nanocomposite are shown in Figure 3. Pure CuO shows ultrathin sheets of varying dimensions ranging from ~50–700 nm in Figure 3a. The sheets across some regions are well stacked together while at other places sheets with fibrillar-like structures are arranged in a haphazard manner, which may be due to breakage or exfoliation of larger sheets [36]. The rGO@CuO nanocomposite in Figure 3b shows a similar morphology to CuO and the sheets of rGO are not clearly observed. This might be attributed to a range of reasons such as their similar morphology to the thin sheets of CuO, rGO sheets covering CuO, or some rGO sheets that might be deeply buried inside the CuO sheet stacks. In the case of Ag-rGO@CuO, from Figure 3c, a large number of small Ag particles in clusters of varying sizes can be seen covering the sheets and stacked between loosely-packed sheets. EDX analysis of the Ag-rGO@CuO nanocomposite in Figure 3d shows only the elemental peaks of C, Cu, O and Ag, which suggests that the nanocomposite is free from impurities and supports the efficacy of the synthesis method (Table 1). Furthermore, EDX-elemental mapping of Ag-rGO@CuO shows the uniform distribution of the respective elements Figure 4a–e.

Transmission electron microscopy was employed to analyze the size and shape of Ag-rGO@CuO nanocomposite at different magnifications and the micrographs are shown in Figure 4. Large clusters of Ag nanoparticles, CuO and rGO sheets of dimensions below 100 nm were observed. Figure 5a represents the low magnification image of Ag-rGO@CuO nanocomposite, which indicates that the CuO nanosheets were grafted by the rGO sheets. At low magnification in Figure 5b, it can be seen that the Ag nanoparticles decorate the rGO@CuO nanocomposite surface. At high magnification, in Figure 5c, the Ag nanoparticles are shown to be 20–50 nm in size.

### 3.3. Raman Spectrum

Raman spectra of the CuO nanosheets and their nanocomposites are shown in Figure 6. All the samples show two common Raman peaks located at 284 and 620 cm^−1^, which are attributed to the A_g_ and B_g_^2^ modes of the monoclinic CuO structure. The slight shifting of peaks with respect to bulk CuO may be due to different nanoscale structure and morphology [37]. Similar shifting was also observed by Murthy and Venugopalan in their nanosized CuO [38]. In the rGO@CuO composite, due to GN, there are two prominent peaks at 1340 and 1593 cm^−1^ called D and G bands, respectively. The D band represents defects or disorder created by the attachment of functional groups containing oxygen and the G band denotes graphitization arising from first-order scattering of the E_2g_ mode [39]. Similarly, the Ag-rGO@CuO composite shows the same characteristic D and G bands as in the rGO@CuO composite but with peaks at different positions: 1352 and 1587 cm^−1^ respectively. The inset of Figure 5 shows the Raman spectrum for GO with peaks at 1360 and 1602 cm^−1^ for D and G band, respectively. The intensity ratio of the D and G bands (I_D_/I_G_) in GO, rGO@CuO, and Ag-rGO@CuO were determined to be 0.90, 1.03, and 1.06, respectively. The ratio of (I_D_/I_G_) present in rGO@CuO, and Ag-rGO@CuO is greater than GO which confirms the reduction of GO to rGO. Similar results were also reported by Mehti et al. [40], where the ratio (I_D_/I_G_) rGO in rGO-ZnO nanocomposite was found to be greater than of GO and was interpreted to predict the reduction of GO into rGO during microwave heating. Apart from this, the NaBH_4_ used in reduction of AgNO_3_ has also been reported to reduce GO into rGO by Shin et al. [41] The higher values of I_D_/I_G_ in rGO@CuO than GO reveal that the number of defects is increased during the reduction of GO into rGO. Further increase in the intensity ratio in Ag-rGO@CuO might be due to an increase in the number of defects after incorporation of Ag nanoparticles on the surface as well as between the sheets. In the Ag-rGO@CuO spectrum, the D and G band intensities were higher than those for rGO@CuO because the Raman signal was increased by surface-enhanced Raman scattering (SERS) [42,43].

### 3.4. Electrochemical Capacitive Performance Analysis

The electrochemical performance of the prepared electrodes was examined by CV and GCD methods using three-electrode cells in a freshly-prepared 2M solution of KOH electrolyte. CV is a very important tool to measure the reduction-oxidation (redox) behavior as well as the capacitive nature of the electrode material. Figure 7a–c represents the cyclic voltammetry curves corresponding to pure CuO nanosheets, rGO@CuO, and Ag-rGO@CuO at different scan rates (10–50 mVs^−1^) and Figure 8a shows the comparative CV curve of all the prepared three electrodes at a fixed scan rate of 50 mVs^−1^ in a potential range of 0.0–0.5 V. The CV curve of each electrode shows the pair of anodic and cathodic redox peaks that appear during electrochemical reactions. The presence of two visible peaks (redox peaks) confirms that all the samples display pseudocapacitive behavior in contrast with an EDLC where the CV profile takes a rectangular form with no obvious redox peaks. In a CuO-based system, oxidation and reduction of electrons between the electrode and electrolyte takes place by faradaic redox reactions as follows:(1)$$CuO+\frac{1}{2}H_{2}O\Leftrightarrow \frac{1}{2}Cu_{2}O+OH^{-}$$

The redox signal of all electrodes is attributed to the oxidation of Cu^+^ to Cu^2+^ and the reduction of Cu^2+^ to Cu^+^ [44]. In Figure 7a the pure CuO nanosheets show asymmetric redox peaks at different scan rates but with a slight shifting observed at a higher scanning rate indicating poor electron mobility at the electrode-electrolyte interface [45].

In the CV curve of the rGO@CuO composite (Figure 7b) both the anodic and cathodic peak curves appear to be higher than those for the CuO sheets because the introduction of rGO sheets into CuO significantly reduces CuO nanosheet aggregation and creates more active sites on the electrode surface for redox reactions to take place. These extra active sites improve the conductivity and capacitance behavior of rGO@CuO by providing new conducting paths for electron transfer. Therefore, the capacitive behavior of rGO@CuO is increased in comparison with pure CuO electrode material. A nanocomposite based on CuO and GN has been reported to show high performance as an anode material for lithium-ion batteries [46]. In the case of Ag-rGO@CuO Figure 7c, the redox peak current is higher than in both pure CuO and the rGO@CuO nanocomposite due to the surface decoration by Ag nanoparticles. As outlined previously, the excellent electrical conductivity of Ag provides more channels for electron transfer during the redox process. The results from the Ag-rGO@CuO nanocomposite demonstrate that this type of material could be an emerging candidate in supercapacitor applications [47]. The CV curves for the pure CuO ultra-fine nanosheets, the rGO@CuO binary and the Ag-rGO@CuO ternary composite at fixed scan rate (50 mVs^−1^) over the potential range 0.0–0.5 V are shown in Figure 8a. Out of all three electrode materials investigated, the ternary nanocomposite Ag-rGO@CuO showed the best electrochemical performance as determined from the high electrochemical surface area enclosed by its CV curve. The specific capacitance of pure CuO and its nanocomposites was calculated by the following formula:(2)$$Q_{s}=\frac{1}{vm}\int_{vi}^{vf}I\times VdV$$
where *Q_s_* represents the specific capacity (Fg^−1^), *V* denotes the scan rate (Vs^−1^), *m* is the mass of active electrode material in grams loaded on Ni foam, and the integrated area represents the anodic peak area under the CV curve. The specific capacitance of Ag-rGO@CuO was found to be 689 Fg^−1^ which is much higher than that for rGO@CuO (511 Fg^−1^) and pure CuO nanosheets (202 Fg^−1^).

Galvanostatic charge-discharge (GCD) was performed to investigate the rate capability of the different electrode materials. The individual GCD curves were recorded at different current densities within a potential difference of 0.4 V as shown in Figure 9a–c. Figure 8b shows the GCD curves of all fabricated electrodes at a fixed current load of 0.5 Ag^−1^. The specific capacitance of the pure CuO sheets (Figure 9a) was 113.75, 112.50, 100, 100 and 87.50 Fg^−1^ at current loads of 0.5, 1, 2, 5 and 7 Ag^−1^, respectively. Similarly, the specific capacitances of the binary rGO@CuO (Figure 9b) and the ternary Ag-rGO@CuO nanocomposite (Figure 9c) were 375, 370, 265, 237.5, 87.50 Fg^−1^ and were 612.5, 605, 405, 300, 210 Fg^−1^ at current loads of 0.5, 1, 2, 5, 7 Ag^−1^, respectively. The Ag-rGO@CuO nanocomposite electrode shows excellent capacitive performance in comparison to rGO@CuO and pure CuO nanosheets. This is due to the addition of rGO which provides more active sites or conducting paths, which increases electron and ion mobility, limiting unwanted aggregation within the CuO and rGO sheets. The silver nanoparticles on the surface of rGO@CuO provide extended channels for electron transfer during the redox process. The specific capacitive value of CuO nanosheets and its nanocomposites was calculated from the GCD curves using the following equation:(3)$$C_{sp}=\frac{Idt}{mdv}$$
where *C_sp_* is the specific capacitance (Fg^−1^), *I* is the charge-discharge current (A), *dt* is the discharge time (s), *m* denotes the mass (g) of the active material loaded onto the Ni foam and *dv* represents the voltage difference between the upper and lower potential.

From Figure 10a the ternary Ag-rGO@CuO nanocomposite shows a very high specific capacitance (612.5 Fg^−1^) at a current load of 0.5 (Ag^−1^) as compared to rGO@CuO (375 Fg^−1^) and pure CuO nanosheets (113.75 Fg^−1^). Again, this enhancement in Ag-rGO@CuO is due to the incorporation of rGO and Ag nanoparticles which create more electrochemically active sites between the electrolyte and electrode.

In addition to studying the capacitive behavior of the prepared electrodes by CV and GCD, the charge-discharge cycling stability was examined. To check the cycle stability, 3000 charge-discharge cycles were performed and the results are shown in Figure 10b. The Ag-rGO@CuO nanocomposite shows excellent cycle stability with up to 92% of the initial capacitance retained. Under similar test conditions the CuO nanosheets retained up to 60% of the initial capacitance. The high capacitance retention in the Ag-rGO@CuO nanocomposite is once more due to the presence of rGO which provides conducting paths for electron flow, and the incorporation of Ag nanoparticles which increases electron mobility and the creation of more active sites during the charge-discharge process. In comparison, the CuO electrode has fewer active sites due to the aggregation of nanosheets, which reduces the overall electron mobility. The capacitance of the prepared ternary nanocomposite (Ag-rGO@CuO) is compared against other reported pure CuO and CuO based nanocomposites in Table 2.

## 4. Conclusions

In this study, pure CuO nanosheets, a binary composite of rGO@CuO and a ternary composite of Ag-rGO@CuO were synthesized by a microwave-assisted hydrothermal and chemical reduction method. The prepared electrode materials were characterized by different analytical techniques to confirm their formation and decoration of Ag nanoparticles upon nanosheets of rGO and CuO. Electrochemical studies showed that the ternary nanocomposite (Ag-rGO@CuO) exhibited a high specific capacitance of 612 Fg^−1^ at 0.5 Ag^−1^ which was higher than that found for pure CuO nanosheets and the rGO@CuO composite. The high capacitive value of the ternary composite was obtained by incorporating rGO thin sheets into ultrafine CuO sheets which provided a conductive platform within the rGO and CuO sheets. Furthermore, the Ag nanoparticles also created new conducting channels which further facilitated ion conduction. The Ag-rGO@CuO composite exhibited very good cycle stability after 3000 cycles with 92% capacitance retention. Thus, an electrode material based on Ag-rGO@CuO ternary nanocomposite may open a gateway to fabricate high-performance electrochemical storage devices.

## Figures and Tables

**Figure 1 materials-14-05032-f001:**
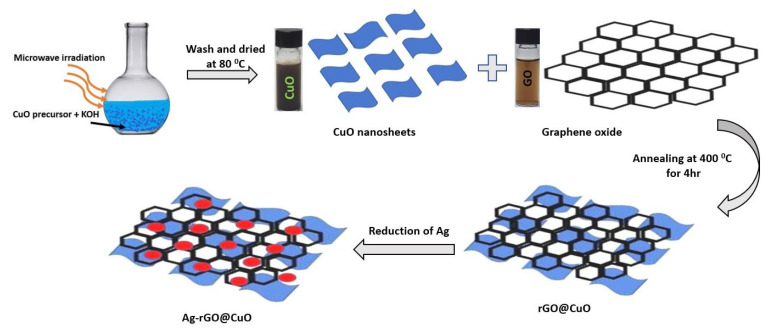
Schematic diagram representing the fabrication of Ag-rGO@CuO nanocomposite.

**Figure 2 materials-14-05032-f002:**
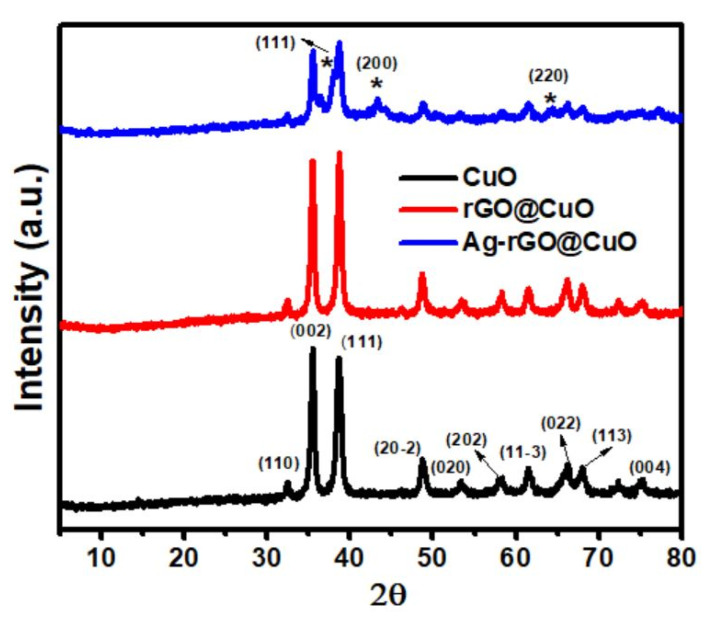
XRD pattern of pure CuO nanosheets, rGO@CuO, and Ag-rGO@CuO nanocomposites. The asterisk (*) represents the peak positions of Ag.

**Figure 3 materials-14-05032-f003:**
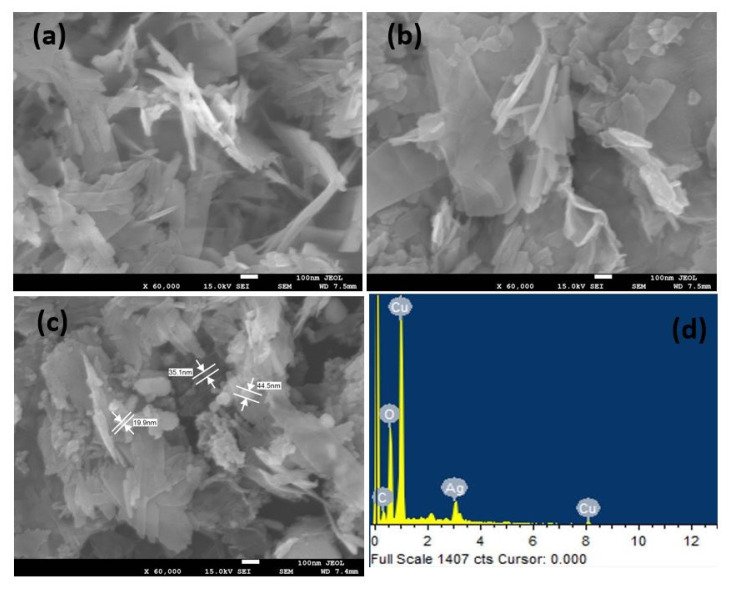
SEM image of (**a**) pure CuO nanosheets, (**b**) rGO@CuO and (**c**) Ag-rGO@CuO. EDX of Ag-rGO@CuO nanocomposite (**d**).

**Figure 4 materials-14-05032-f004:**
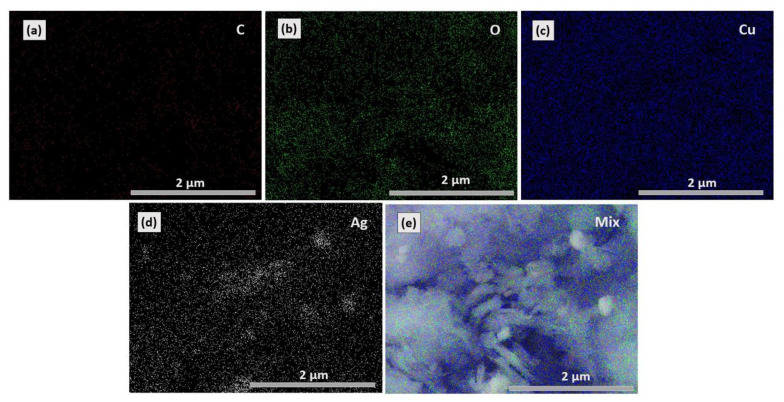
Elemental mapping of (**a**) C, (**b**) O, (**c**) Cu and (**d**) Ag content present in Ag-rGO@CuO nanocomposite. The Ag-rGO@CuO nanocomposite shows uniform distribution of all elements i.e., C, O, Cu and Ag (**e**).

**Figure 5 materials-14-05032-f005:**
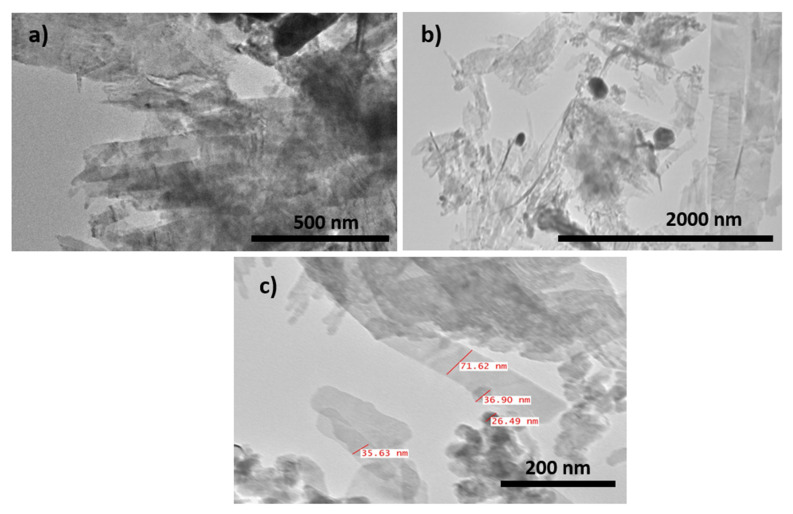
TEM image of Ag-rGO@CuO nanocomposite at different magnifications (**a**) 10,000× (**b**) 15,000× and (**c**) 80,000×.

**Figure 6 materials-14-05032-f006:**
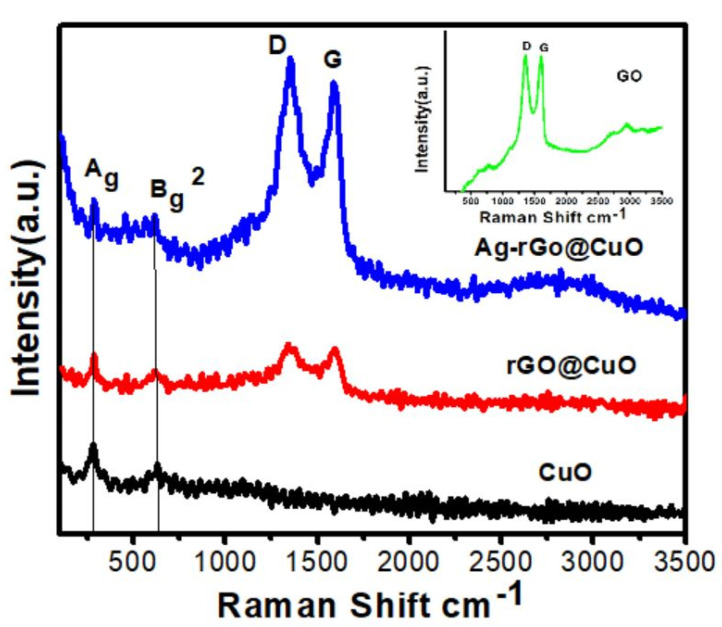
Raman shift of pure CuO nanosheets, rGO@CuO and Ag-rGO@CuO nanocomposites.

**Figure 7 materials-14-05032-f007:**
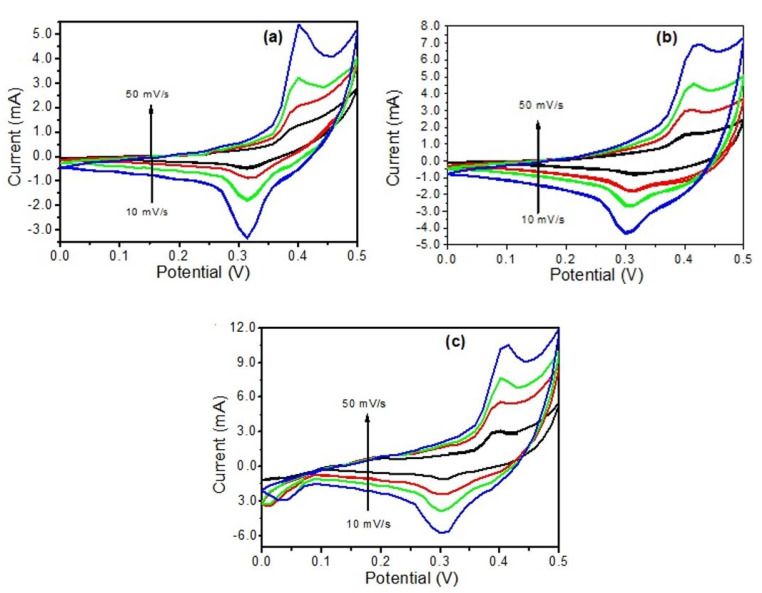
CV curve of (**a**) pure CuO nanosheets, (**b**) rGO@CuO and (**c**) Ag-rGO@CuO nanocomposites at different scan rates.

**Figure 8 materials-14-05032-f008:**
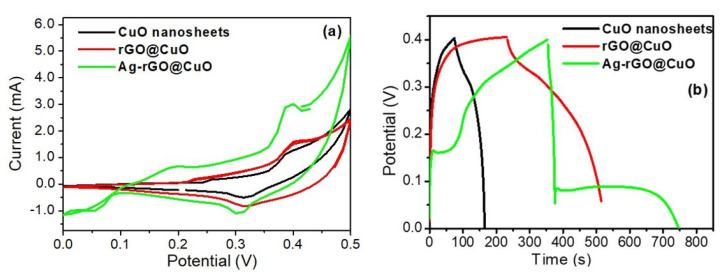
Comparative (**a**) cyclic voltammetry (CV) and (**b**) GCD plots of pure CuO nanosheets, rGO@CuO, and Ag-rGO@CuO.

**Figure 9 materials-14-05032-f009:**
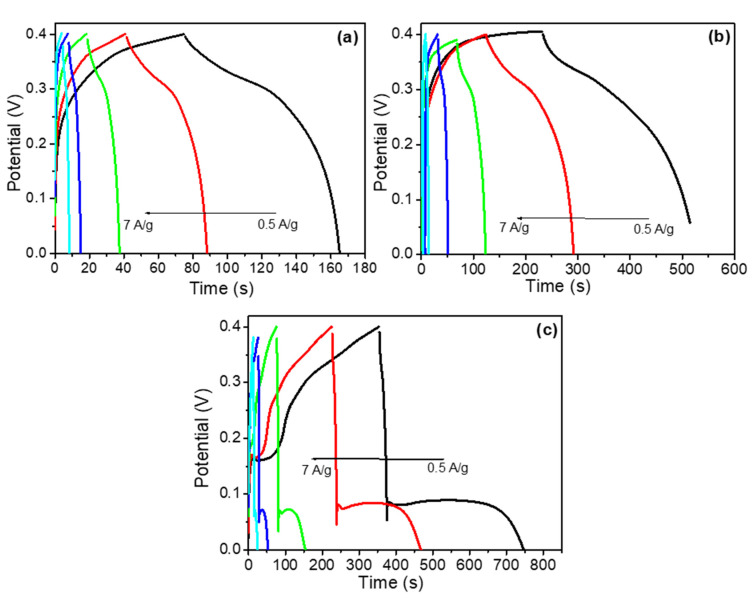
(**a**) GCD curve of pure CuO nanosheets, (**b**) rGO@CuO and (**c**) Ag-rGO@CuO nanocomposites at different current loads.

**Figure 10 materials-14-05032-f010:**
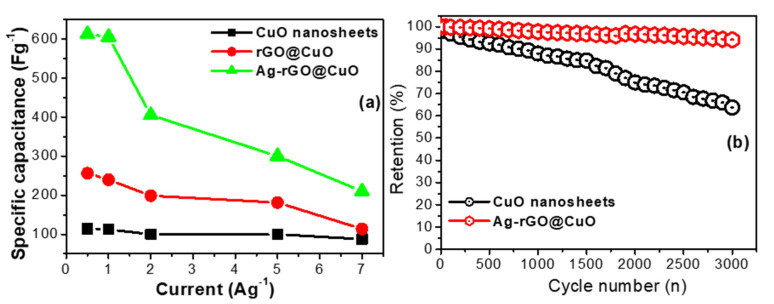
(**a**) Calculated specific capacitance of pure CuO nanosheets, rGO@CuO, and Ag-rGO@CuO at different current densities, (**b**) cycling stability graph of the CuO nanosheets and Ag-rGO@CuO nanocomposite.

**Table 1 materials-14-05032-t001:** EDX analysis of ternary Ag-rGO@CuO nanocomposite.

Element	Weight%	Atomic%
C K	4.15	12.64
O K	20.82	47.63
Cu L	60.32	34.74
Ag L	14.71	4.99

**Table 2 materials-14-05032-t002:** Comparison between the capacitance of prepared ternary nanocomposite Ag-rGO@CuO against reported pure CuO and CuO based nanocomposites.

Electrode Material	Electrolyte	Specific Capacitance (C_sp_) Fg^−1^	Current Density, Ag^−1^	No. of Cycles	Retention %	Ref.
CuO NR_r_	1M Na_2_ SO_4_	206.6	1	1000	88	[48]
CuO Cauliflower	1M Na_2_ SO_4_	179	2 mAcm^−2^	2000	81	[10]
CuO/PANI	1M Na_2_ SO_4_	185	5 mVs^−1^	2000	72	[49]
rGO/Cu_2_O	1M KOH	98	1	1000	50	[50]
GO/CuO	1M Na_2_ SO_4_	245	0.1	1000	79	[51]
CuO/N-rGO	6M KOH	340	0.5	500	80	[52]
Ag-rGO@CuO	2M KOH	612.5	0.5	3000	92	Present case

## Data Availability

Not Applicable.

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
