# Peer review of "Silver Nanoparticles Embedded on Reduced Graphene Oxide@Copper Oxide Nanocomposite for High Performance Supercapacitor Applications"

_materials, 2021, doi:10.3390/ma14175032_

Round 1
Reviewer 1 Report
This manuscript studied ternary nanocomposites for supercapacitors. The nanocomposites are synthesized by the traditional hydrothermal method and Hummer's Method with chemical reduced Ag Nps. the author claimed that the synergic effect among them and boosted the electrochemical performance. However, there is limited evidence to support the conclusion clearly and some figures must be improved.
- Please confirm the contribution of rGO in your final composite by more characterization as well as its content.
- The Ag should be well marked in XRD.
- Please explain why the intensity of CuO decreases in CuO-rGO@A
- EIS should be provided to confirm the improvement of conductivity
- The sampling rate affects the figures of electrochemical evaluation, resulting in poor quality of plotting lines.
- The mass loading should be provided.
Reviewer 2 Report
The introduction starts with a minor error - two instances of the word "The" in the first line.
The authors obviously think that their material could be used in supercapacitors so perhaps this should be reflected in the article title?
The abstract is clear and the introduction is good.
The materials and methods is reasonably clearly described. However, figure 1 needs to have some supporting photos such as a pictures of the GO suspension and also the CuO nanosheet suspension.
In line 140 the authors sate that the nanocomposite of GO@CuO was annealed at 400C for 4 hrs to reduce the GO to rGO. This needs to be explained and shown that the GO has actually been reduced.
A picture of the Ag-rGO@CuO composite material. Line 149 say this is Ag-rGO@ZnO but the authors must mean Cu and typed Zinc by mistake? A picture of the electrochemical cell would also be useful.
In the results and discussion section the results are presented but not really discussed properly. The authors could have a look at the results and discussion section of Ref 51 and follow their example?
The XRD data needs to start at 5 degrees 2 Theta and XRD patterns for the GO material and for the rGO material is needed. From the XRD data and the Raman data it is not clear that the authors have rGO material.
The SEM micrographs and presentation of the EDX data needs improvement. In line 219, the authors say that the Ag nanoparticles are ca. 20-50n nm in size so this should be clear in the SEM micrograph. The authors could lok at rhe TEM grid for figure 4 (c) in SEM?
The EDX data should be put in a table to show elements present and amount. If possible, it would help clarity if the authors could perform SEM/EDX mapping to show the Cu and Ag on the nanocomposite?
Round 2
Reviewer 2 Report
Thank you for making the timely correction to your manuscript.